DATA RELEASE

# American triatomine species occurrences: updates and novelties in the DataTri database

Soledad Ceccarelli[1,2,*], Agustín Balsalobre[1,2], María Eugenia Vicente[1], Rachel Curtis-Robles[3], Sarah A. Hamer[3], José Manuel Ayala Landa[4], Jorge E. Rabinovich[1,2] and Gerardo A. Marti[1,2]

1 Centro de Estudios Parasitológicos y de Vectores (CEPAVE-CCT-La Plata-CONICET-UNLP), La Plata, Buenos Aires 1900, Argentina
2 Consejo Nacional de Investigaciones Científicas y Técnicas (CONICET), Buenos Aires, Argentina
3 College of Veterinary Medicine and Biomedical Sciences, Texas A&M University, College Station, Texas, USA
4 Museo del Instituto de Zoología Agrícola (MIZA), Maracay, Venezuela

## ABSTRACT

The causative agent of Chagas disease (*Trypanosoma cruzi*) is transmitted to mammals, including humans, mainly by insect vectors of the subfamily Triatominae (Hemiptera: Reduviidae). Also known as "kissing bugs", the subfamily currently includes 157 validated species (154 extant and three extinct), in 18 genera and five tribes. Here, we present a subdataset (7852 records) of American triatomine occurrences; an update to the most complete and integrated database available to date at a continental scale. New georeferenced records were obtained from a systematic review of published literature and colleague-provided data. New data correspond to 101 species and 14 genera from 22 American countries between 1935 and 2022. The most important novelties refer to (i) the inclusion of new species, (ii) synonymies and formal transferals of species, and (iii) temporal and geographical species records updates. These data will be a useful contribution to entomological surveillance implicated in Chagas disease.

**Subjects**  Animal and Plant Sciences, Biodiversity, Ecology

**Submitted:** 29 March 2022

\* Corresponding author. E-mail: soledad.ceccarelli@gmail.com

Preprint submitted at https://doi.org/10.1590/SciELOPreprints.4151

Included in the series: ***Vectors of human disease*** (https://doi.org/10.46471/GIGABYTE_SERIES_0002)

## DATA DESCRIPTION

### Context

Chagas disease, caused by the protozoan *Trypanosoma cruzi* (Chagas, 1909) (NCBI:txid5693) (Kinetoplastida, Trypanosomatidae), is transmitted mainly through the feces of triatomine (Hemiptera: Reduviidae: Triatominae) insect vectors, but may also be transmitted from mother to child, by blood transfusions or some infected organ transplants, and by oral transmission through contaminated food and/or beverages. These multiple routes of transmission make Chagas disease an important public health problem, primarily in the Americas but also on other continents [1]. Currently, the subfamily Triatominae consists of 154 extant and three fossil species (143 species present in America and 14 species present in Asia), grouped into five tribes and 18 genera [2]. Assembling geographic and ecological information about triatomines is particularly important, given that these insect vectors are one of the main routes of *T. cruzi* transmission. Once available, such information may be used by public health agencies to carry out actions and support programs for disease prevention and vector control.

Since the publication on American triatomine species by Carcavallo *et al.* [3], a new, complete and integrated database on triatomine occurrences, called DataTri [4] has been available. However, as the number of species has increased and there have been several changes in triatomine taxonomy, this information needs an update. In this context, the main goal of this work is to describe the features of a subdataset of American triatomine occurrences (henceforth referred to as the "American subdataset"), highlighting the most important updates and inclusions after 5 years spent reviewing new data sources. This American subdataset was integrated to update the current American triatomine occurrence dataset (henceforth referred to as the "New American dataset"), comprising 19,600 records and available via the Global Biodiversity Information Facility (GBIF) platform [5]. In turn, this integrated New American dataset is complemented by a dataset on triatomine species present in Argentina (henceforth referred to as the "Argentinean dataset"), also stored in GBIF (Figure 1).

This work is the result of an exhaustive review of public information combined with substantial interinstitutional collaboration, which integrated not only geographical but also ecological data for all American triatomine species spanning 24 American countries. This geodatabase may contribute not only towards improving knowledge of the geographical distributions of every American triatomine species, but also to designing improved strategies for health promotion and vector control. We believe it will be of practical use for both the academic and educational community, as well as for those institutions responsible for public health promotion, prevention and vector control activities.

## Triatomine species occurrence datasets: background

In 2018, an initial dataset (comprising 21,815 georeferenced records) of triatomine species occurrence (called DataTri) was created, described [4] and stored in Figshare [6]. Subsequently, in 2020, another 5893 records were incorporated (amounting to 27,708 records). Then, it was modified to Darwin Core format [7] and stored in the GBIF platform, split into two complementary datasets: the "American dataset" (11,791 records, spanning 1926–2022) and the "Argentinean dataset" (15,917 records, spanning 1912–2019) [8] (Figure 1a). Both datasets are complementary because the American dataset includes all American triatomine species except for the 17 species (namely *Panstrongylus geniculatus* (Latreille, 1811), *P. guentheri* Berg, 1879, *P. megistus* (Burmeister, 1835), *P. rufotuberculatus* (Champion, 1899), *Psammolestes coreodes* Bergroth, 1911, *Triatoma breyeri* Del Ponte, 1929, *T. delpontei* Romaña & Abalos, 1947, *T. eratyrusiformis* Del Ponte, 1929, *T. garciabesi* Carcavallo, Cichero, Martínez, Prosen & Ronderos, 1967, *T. guasayana* Wygodzinsky & Abalos, 1949, *T. infestans* (Klug, 1834), *T. limai* Del Ponte, 1929, *T. patagonica* Del Ponte, 1929, *T. platensis* Neiva, 1913, *T. rubrofasciata* (De Geer, 1773), *T. rubrovaria* (Blanchard, 1843) and *T. sordida* (Stål, 1859)) that are included in the Argentinean dataset and recorded as having (or having had) at least part of their geographical distribution within Argentine territory.

## New American triatomine occurrence data

To build a new subdataset of American triatomine occurrences (the American subdataset) and integrate it into the New American dataset available at present, a total of 7852 occurrence records were compiled from 2 years (2020–2022) of systematic reviews (Figure 1b, Table 1). The new records (*N* = 7852) are identified consecutively from the "catalogNumber = 27631" to the "catalogNumber = 35482" in the New American dataset.



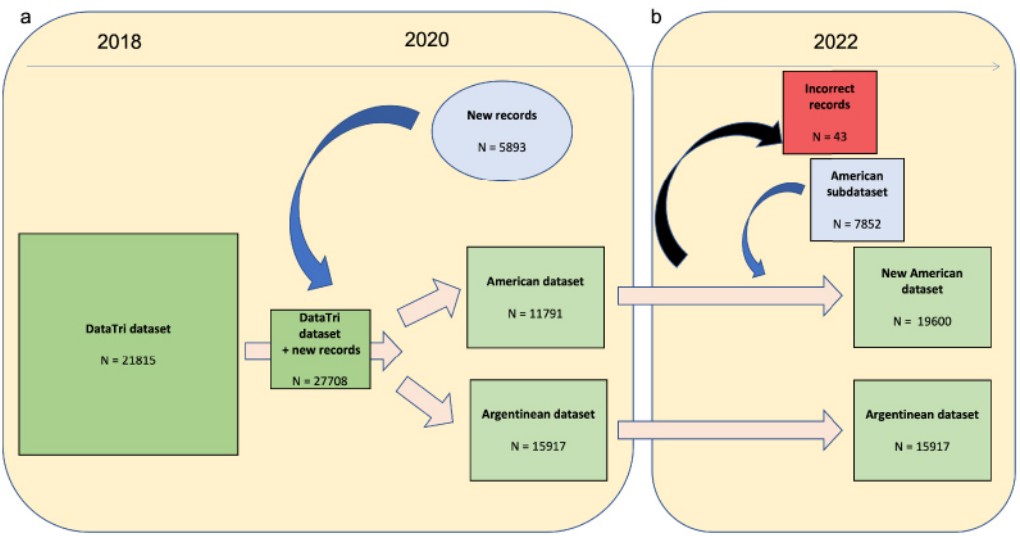

**Figure 1.** Schematic overview of the conformation of triatomine datasets in a temporal frame. *N* = number of records in each dataset/subdataset. (a) Actions done between 2018 and 2020, and (b) actions done in 2022. The black arrow indicates elimination of incorrect records (repeated records). The blue arrow indicates addition of new records.

The most important novelties of the American subdataset are: (i) the incorporation of new triatomine species described between 2018 and 2021, as well as species (extant and extinct) not originally included in the DataTri dataset; (ii) synonymies and formal transferals of species; and (iii) new records that update several temporal and geographical species occurrences.

Seven new extant species were incorporated: *Bolbodera scabrosa* (Valdés, 1910), *Belminus santosmalletae* (Dale, Justi & Galvão, 2021), *Rhodnius micki* (Zhao, Galvão & Cai, 2021), *Nesotriatoma obscura* (Maldonado & Farr, 1962), *Nesotriatoma confusa* (de Oliveira, Ayala, Justi, da Rosa & Galvão, 2018), *Triatoma huehuetenanguensis* (Lima-Cordón, Monroy, Stevens, Rodas, Rodas, Dorn & Justi, 2019) and *T. mopan* (Dorn, Justi, Dale, Stevens, Galvão, Lima-Cordón & Carlota Monroy, 2018). The two extinct (†) ones were *Triatoma dominicana* (Poinar, 2005)† and *Panstrongylus hispaniolae* (Ponair Jr., 2013)†. With these incorporations, the New American dataset includes 125 species that, together with the 17 species included in the Argentinean dataset, make a total of 142 triatomine species. Except for the particular case of the species *T. rosai* (Alevi, de Oliveira, Caris Garcia, Cesaretto Cristal, Grzyb Delgado, de Freitas Bittinelli, Visinho dos Reis, Ravazi, Bortolozo de Oliveira, Galvão, Vilela de Azeredo-Oliveira & Fernandez Madeira, 2020), which was described in 2020 after the last update of the Argentinean dataset (and therefore not included in the latter), occurrence records of all the American triatomine species described up to date [2] are included between both datasets (Table 1).

Taxonomic updates were: (i) synonymities: *Rhodnius taquarussuensis*, which was described through integrative taxonomy [9] was later synonymized with *R. neglectus* (Lent, 1954) [10]; some records of *Nesotriatoma bruneri* were synonymized with *Nesotriatoma flavida* (Neiva, 1911) [11]; some records of *Triatoma dimidiata* from Guatemala, Honduras and México were synonymized with *T. huehuetenanguensis* [12]; and (ii) formal transferals: *Triatoma flavida* was transferred into the genus *Nesotriatoma* with the resulting new



**Table 1.** Current taxa classification of the extant and extinct (†) American triatomine species according to the last taxonomic classification of Alevi *et al.* (2021) [2]. The "x" indicates the triatomine species included in the New American dataset or in the Argentinean dataset.

| Tribes | Genera | Species name (descriptor name(s), year) | Species included in the New American dataset | Species included in the Argentinean dataset | Number of records in the New American dataset | Number of records contributed by the American subdataset |
|---|---|---|---|---|---|---|
| Alberproseniini | *Alberprosenia* | *goyovargasi* (Martínez & Carcavallo, 1977) | x | - | 1 | 1 |
| | | *malheiroi* (Serra, Atzingen & Serra, 1980) | x | - | 3 | 0 |
| Bolboderini | *Belminus* | *corredori* (Galvão & Angulo, 2006) | x | - | 1 | 0 |
| | | *costaricensis* (Herrer, Lent & Wygodzinsky, 1954) | x | - | 2 | 0 |
| | | *ferroae* (Sandoval, Pabón, Jurberg & Galvão, 2007) | x | - | 4 | 0 |
| | | *herreri* (Lent & Wygodzinsky, 1979) | x | - | 22 | 6 |
| | | *laportei* (Lent, Jurberg & Carcavallo, 1995) | x | - | 3 | 1 |
| | | *peruvianus* (Herrer, Lent & Wygodzinsky, 1954) | x | - | 6 | 0 |
| | | *pittieri* (Osuna & Ayala, 1993) | x | - | 1 | 0 |
| | | *rugulosus* (Stål, 1859) | x | - | 2 | 1 |
| | | *santosmalletae* (Dale, Justi & Galvão, 2021) | x | - | 0 | 1 |
| | *Bolbodera* | *scabrosa* (Valdés, 1910) | x | - | 0 | 3 |
| | *Microtriatoma* | *borbai* (Lent & Wygodzinsky, 1979) | x | - | 5 | 1 |
| | | *trinidadensis* (Lent, 1951) | x | - | 12 | 1 |
| | *Parabelminus* | *carioca* (Lent, 1943) | x | - | 1 | 0 |
| | | *yurupucu* (Lent & Wygodzinsky, 1979) | x | - | 2 | 0 |
| Cavernicolini | *Cavernicola* | *lenti* (Barrett & Arias, 1985) | x | - | 3 | 0 |
| | | *pilosa* (Barber, 1937) | x | - | 39 | 3 |
| Rhodniini | *Psammolestes* | *arthuri* (Pinto, 1926) *pilosa* (Barber, 1937) | x | - | 15 | 6 |
| | | *coreodes* (Bergroth, 1911) *pilosa* (Barber, 1937) | - | x | 0 | 0 |
| | | *tertius* (Lent & Jurberg, 1965) | x | - | 91 | 17 |
| | *Rhodnius* | *amazonicus* (Almeida, Santos & Sposina, 1973) | x | - | 6 | 2 |
| | | *barretti* (Abad-Franch, Pavan, Jaramillo, Palomeque, Dale, Chaverra & Monteiro, 2013) | x | - | 2 | 1 |
| | | *brethesi* (Matta, 1919) | x | - | 12 | 7 |
| | | *colombiensis* (Mejia, Galvão & Jurberg, 1999) | x | - | 23 | 1 |
| | | *dalessandroi* (Carcavallo & Barreto, 1976) | x | - | 2 | 0 |
| | | *domesticus* (Neiva & Pinto, 1923) | x | - | 17 | 2 |
| | | *ecuadoriensis ecuadoriensis* (Lent & León, 1958) | x | - | 44 | 64 |
| | | *marabaensis* (dos Santos Souza, Barbosa Von Atzingen, Furtado, de Oliveira, Damieli Nascimento, Pagotto Vendrami & Aristeu da Rosa, 2016) | x | - | 1 | 0 |
| | | *micki* (Zhao, Galvão & Cai, 2021) | x | - | 0 | 2 |
| | | *milesi* (Carcavallo, Rocha, Galvão & Jurberg, 2001 (in: Valente *et al.* 2001) | x | - | 2 | 1 |
| | | *montenegrensis* (Rosa, Rocha, Gardim, Pinto, Mendonça, Ferreira Filho, Carvalho, Camargo, Oliveira, Nascimento, Cilense & Almeida, 2012) | x | - | 3 | 5 |
| | | *nasutus* (Stål, 1859) | x | - | 66 | 2 |
| | | *neglectus* (Lent, 1954) | x | - | 139 | 9 |
| | | *neivai* (Lent, 1953) | x | - | 6 | 1 |
| | | *pallescens* (Barber, 1932) | x | - | 126 | 41 |
| | | *paraensis* (Sherlock, Guitton & Miles, 1977) | x | - | 7 | 2 |
| | | *pictipes* (Stål, 1872) | x | - | 164 | 16 |
| | | *prolixus* (Stål, 1859) | x | - | 1214 | 16 |
| | | *robustus* (Larrousse, 1927) | x | - | 115 | 6 |
| | | *stali* (Lent, Jurberg & Galvão, 1993) | x | - | 10 | 2 |
| | | *zeledoni* (Jurberg, Rocha & Galvão, 2009) | x | - | 1 | 0 |
| Triatomini | *Dipetalogaster* | *maxima* (Uhler, 1894) | x | - | 15 | 28 |
| | *Eratyrus* | *cuspidatus* (Stål, 1859) | x | - | 59 | 45 |
| | | *mucronatus* (Stål, 1859) | x | - | 91 | 14 |
| | *Hermanlentia* | *matsunoi* (Fernández-Loayza, 1989) | x | - | 4 | 0 |

**Table 1.** (Continued)

| Tribes | Genera | Species name (descriptor name(s), year) | Species included in the New American dataset | Species included in the Argentinean dataset | Number of records in the New American dataset | Number of records contributed by the American subdataset |
|---|---|---|---|---|---|---|
| | *Mepraia* | *gajardoi* (Frias, Henry & Gonzalez, 1998) | x | - | 11 | 17 |
| | | *parapatrica* (Frias-Lasserre, 2010) | x | - | 6 | 6 |
| | | *spinolai* (Porter, 1934) | x | - | 24 | 152 |
| | *Nesotriatoma* | *confusa* (de Oliveira, Ayala, Justi, da Rosa & Galvão, 2018) | x | - | 2 | 11 |
| | | *flavida* (Neiva, 1911) | x | - | 7 | 1 |
| | | *obscura* (Maldonado & Farr, 1962) | x | - | 0 | 3 |
| | *Panstrongylus* | *chinai* (Del Ponte, 1929) | x | - | 64 | 16 |
| | | *diasi* (Pinto & Lent, 1946) *chinai* (Del Ponte, 1929) | x | - | 24 | 8 |
| | | *geniculatus* (Latreille, 1811) | - | x | 0 | 0 |
| | | *guentheri* (Berg, 1879) | - | x | 0 | 0 |
| | | *howardi* (Neiva, 1911) | x | - | 3 | 6 |
| | | *humeralis* (Usinger, 1939) | x | - | 9 | 0 |
| | | *lenti* (Galvão & Palma, 1968) | x | - | 2 | 0 |
| | | *lignarius* (Walker, 1873) | x | - | 53 | 9 |
| | | *lutzi* (Neiva & Pinto, 1923) | x | - | 614 | 41 |
| | | *martinezorum* (Ayala, 2009) | x | - | 3 | 0 |
| | | *megistus* (Burmeister, 1835) | - | x | 0 | 0 |
| | | *mitarakaensis* (Bérenger & Blanchet, 2007) | x | - | 1 | 0 |
| | | *rufotuberculatus* (Champion, 1899) | - | x | 0 | 0 |
| | | *tupynambai* (Lent, 1942) | x | - | 16 | 1 |
| | | *hispaniolae* (Ponair Jr., 2013) † | x | - | 0 | 1 |
| | *Paratriatoma* | *hirsuta* (Barber, 1938) | x | - | 58 | 3 |
| | | *lecticularia* (Stål, 1859) | x | - | 30 | 200 |
| | *Triatoma* | *amicitiae* (Lent, 1951) | - | - | 0 | 0 |
| | | *arthurneivai* (Lent & Martins, 1940) | x | - | 8 | 1 |
| | | *bahiensis* (Sherlock & Serafim, 1967) | x | - | 1 | 2 |
| | | *baratai* (Carcavallo & Jurberg, 2000) | x | - | 11 | 5 |
| | | *barberi* (Usinger, 1939) | x | - | 566 | 32 |
| | | *Bassolsae* (Alejandre Aguilar, Nogueda Torres, Cortéz Jimenez, Jurberg, Galvão & Carcavallo, 1999) | x | - | 2 | 1 |
| | | *bolivari* (Carcavallo, Martínez & Pelaez, 1987) | x | - | 11 | 2 |
| | | *boliviana* (Martínez Avendaño, Chávez Espada,Gil, Aranda Asturizaga, Vargas Mamani & Vidaurre Prieto, 2007) | x | - | 1 | 1 |
| | | *bouvieri* (Larrousse, 1924) | - | - | 0 | 0 |
| | | *brailovskyi* (Martínez, Carcavallo & Pelaez, 1984) | x | - | 16 | 0 |
| | | *brasiliensis* (Neiva, 1911) | x | - | 1042 | 77 |
| | | *breyeri* (Del Ponte, 1929) | - | x | 0 | 0 |
| | | *carcavalloi* (Jurberg, Rocha & Lent, 1998) | x | - | 15 | 1 |
| | | *carrioni* (Larrousse, 1926) | x | - | 18 | 20 |
| | | *cavernicola* (Else & Cheong, 1977) | - | - | 0 | 0 |
| | | *circummaculata* (Stål, 1859) | x | - | 29 | 8 |
| | | *costalimai* (Verano & Galvão, 1958) | x | - | 44 | 2 |
| | | *deaneorum* (Galvão, Souza & Lima, 1967) | x | - | 3 | 0 |
| | | *delpontei* (Romaña & Abalos, 1947) | - | x | 0 | 0 |
| | | *dimidiata* (Latreille, 1811) | x | - | 2102 | 549 |
| | | *dispar* (Lent, 1950) | x | - | 20 | 245 |
| | | *eratyrusiformis* (Del Ponte, 1929) | - | x | 0 | 0 |
| | | *garciabesi* (Carcavallo, Cichero, Martínez, Prosen & Ronderos, 1967) | - | x | 0 | 0 |
| | | *gerstaeckeri* (Stål, 1859) | x | - | 388 | 3740 |
| | | *gomeznunezi* (Martínez, Carcavallo & Jurberg, 1994) | x | - | 2 | 0 |
| | | *guasayana* (Wygodzinsky & Abalos, 1949) | - | x | 0 | 0 |
| | | *guazu* (Lent & Wygodzinsky, 1979) | x | - | 5 | 4 |

**Table 1.** (Continued)

| Tribes | Genera | Species name (descriptor name(s), year) | Species included in the New American dataset | Species included in the Argentinean dataset | Number of records in the New American dataset | Number of records contributed by the American subdataset |
|---|---|---|---|---|---|---|
| | | *hegneri* (Mazzotti, 1940) | x | - | 9 | 2 |
| | | *huehuetenanguensis* (Lima-Cordón, Monroy, Stevens, Rodas, Rodas, Dorn & Justi, 2019) | x | - | 10 | 39 |
| | | *incrassata* (Usinger, 1939) | x | - | 4 | 0 |
| | | *indictiva* (Neiva, 1912) | x | - | 5 | 192 |
| | | *infestans* (Klug, 1834) | - | x | 0 | 0 |
| | | *jatai* (Gonçalves, Teves-Neves, Santos-Mallet, Carbajal de la Fuente, Lopes, 2013) | x | - | 2 | 2 |
| | | *juazeirensis* (Costa y Felix, 2007) | x | - | 17 | 5 |
| | | *jurbergi* (Carcavallo, Galvão & Lent, 1998) | x | - | 8 | 2 |
| | | *klugi* (Carcavallo, Jurberg, Lent & Galvão, 2001) | x | - | 3 | 3 |
| | | *lenti* (Sherlock & Serafim, 1967) | x | - | 16 | 9 |
| | | *leopoldi* (Schoudeten, 1933) | - | - | 0 | 0 |
| | | *limai* (Del Ponte, 1929) | - | x | 0 | 0 |
| | | *longipennis* (Usinger, 1939) | x | - | 381 | 3 |
| | | *maculata* (Erichson, 1848) | x | - | 78 | 96 |
| | | *matogrossensis* (Leite & Barbosa, 1953) | x | - | 11 | 5 |
| | | *mazzottii* (Usinger, 1941) | x | - | 121 | 30 |
| | | *melanica* (Neiva & Lent, 1941) | x | - | 22 | 2 |
| | | *melanocephala* (Neiva & Pinto, 1923) | x | - | 93 | 43 |
| | | *mexicana (Herrich-Schaeffer, 1848)* | x | - | 582 | 36 |
| | | *migrans* (Breddin, 1903) | - | - | 0 | 0 |
| | | *mopan* (Dorn, Justi, Dale, Stevens, Galvão, Lima-Cordón & Carlota Monroy, 2018) | x | - | 0 | 7 |
| | | *neotomae* (Neiva, 1911) | x | - | 10 | 4 |
| | | *nigromaculata* (Stål, 1872) | x | - | 10 | 5 |
| | | *nitida* (Usinger, 1939) | x | - | 50 | 27 |
| | | *oliveirai* (Neiva, Pinto & Lent, 1939) | x | - | 7 | 1 |
| | | *pallidipennis* (Stål, 1872) | x | - | 320 | 8 |
| | | *patagonica* (Del Ponte, 1929) | - | x | 0 | 0 |
| | | *peninsularis* (Usinger, 1940) | x | - | 8 | 0 |
| | | *petrochiae* (Pinto & Barreto, 1925) | x | - | 28 | 4 |
| | | *phyllosoma* (Burmeister, 1835) | x | - | 38 | 26 |
| | | *picturata* (Usinger, 1939) | x | - | 27 | 0 |
| | | *pintodiasi* (Jurberg, Cunha, Cailleaux, Raigorodschi, Souza Lima, da Silva Roch, Figueiredo Moreira, 2013) | x | - | 1 | 0 |
| | | *platensis* (Neiva, 1913) | - | x | 0 | 0 |
| | | *protracta* (Uhler, 1894) | x | - | 341 | 207 |
| | | *pseudomaculata* (Corrêa & Espínola, 1964) | x | - | 1392 | 24 |
| | | *pugasi* (Lent, 1953) | - | - | 0 | 0 |
| | | *recurva* (Stål, 1868) | x | - | 31 | 12 |
| | | *rosai* (Alevi, de Oliveira, Caris Garcia, Cesaretto Cristal, Grzyb Delgado, de Freitas Bittinelli,Visinho dos Reis, Ravazi, Bortolozo de Oliveira, Galvão, Vilela de Azeredo-Oliveira & Fernandez Madeira, 2020) | - | - | 0 | 0 |
| | | *rubida* (Uhler, 1894) | x | - | 158 | 515 |
| | | *rubrofasciata* (De Geer, 1773) | - | x | 0 | 0 |
| | | *rubrovaria* (Blanchard, 1843) | - | x | 0 | 0 |
| | | *ryckmani* (Zeledón & Ponce, 1972) | x | - | 3 | 3 |
| | | *sanguisuga* (Leconte, 1855) | x | - | 124 | 1006 |
| | | *sherlocki* (Papa, Jurberg, Carcavallo, Cerqueira & Barata, 2002) | x | - | 13 | 8 |
| | | *sinaloensis* (Ryckman, 1962) | x | - | 5 | 1 |



**Table 1.** (Continued)

| Tribes | Genera | Species name (descriptor name(s), year) | Species included in the New American dataset | Species included in the Argentinean dataset | Number of records in the New American dataset | Number of records contributed by the American subdataset |
|---|---|---|---|---|---|---|
| | | *sinica* (Hsiao, 1965) | - | - | 0 | 0 |
| | | *sordida* (Stål, 1859) | - | x | 0 | 0 |
| | | *tibiamaculata* (Pinto, 1926) | x | - | 69 | 19 |
| | | *vandae* (Carcavallo, Jurberg, Rocha, Galvão, Noireau & Lent, 2002) | x | - | 4 | 6 |
| | | *venosa* (Stål, 1872) | x | - | 50 | 12 |
| | | *vitticeps* (Stål, 1859) | x | - | 114 | 2 |
| | | *williami* (Galvão, Souza & Lima, 1965) | x | - | 22 | 10 |
| | | *wygodzinskyi* (Lent 1951) | x | - | 8 | 2 |
| | | *dominicana* (Poinar, 2005) † | x | - | 0 | 1 |
| | Linshcosteus | *carnifex* (Distant, 1904) | - | - | 0 | 0 |
| | | *chota* (Lent & Wygodzinsky, 1979) | - | - | 0 | 0 |
| | | *confumus* (Ghauri, 1976) | - | - | 0 | 0 |
| | | *costalis* (Ghauri, 1976) | - | - | 0 | 0 |
| | | *kali* (Lent & Wygodzinsky, 1979) | - | - | 0 | 0 |
| | | *karupus* (Galvão, Patterson, Rocha & Jurberg, 2002) | - | - | 0 | 0 |
| | Paleotriatoma | *metaxytaxa* (Poinar, 2019) † | - | - | 0 | 0 |
| | | **Total number of records** | | | **11,748** | **7852** |

combination, *Nesotriatoma flavida* [11] and *Triatoma lecticularia* was transferred into the genus *Paratriatoma* with the resulting new combination *Paratriatoma lecticularia* (Stål, 1859) [13].

The incorporation of new records from the American subdataset contributes new information (compared with the DataTri dataset) and/or updates the geographic patterns of several species or subspecies in some countries. These are the cases of new occurrences of *Triatoma mopan* and *T. huehuetenanguensis* in Belize [14–16] or *T. dispar* (Lent, 1950), *T. nitida* (Usinger, 1939), *T. ryckmani* (Zeledón & Ponce, 1972), *Cavernicola pilosa* (Barber, 1937), *Eratyrus cuspidatus* (Stål, 1859), *Belminus costaricensis* (Herrer, Lent & Wygodzinsky, 1954) and *Microtriatoma trinidadensis* (Lent, 1951) in Costa Rica [17]. Other cases worth mentioning are *Triatoma sanguisuga* (Leconte, 1855) in Nebraska, USA [18], *Triatoma protracta woodi* (Uhler, 1894), which until now had only been recorded in the wild [19] and now has been recorded in domiciliary habitats in a Texas county [20] and in peridomiciles in New Mexico county [21], and *T. rubida uhleri* (Uhler, 1894), which has records in domiciliary habitats and one record in wild habitat [22]. In French Guyana, records from *Belminus laportei* (Lent, Jurberg & Carcavallo, 1995) were added [23] increasing the number of species represented in this country to 11. Mexico is another example, with records of *Triatoma rubida sonoriana* being a new inclusion for all habitat types (domicile, peridomicile and sylvatic) [24]; new records of *Triatoma barberi* (Usinger, 1939) for Mexico state [25], as well as records of *Triatoma rubida* for Ocampo municipality and records of *Triatoma gerstaeckeri* (Stål, 1859) [26]. Records of some species were also updated in new states of Brazil, such as *Eratyrus mucronatus* (Stål, 1859) and *Rhodnius montenegrensis* (Rosa, Rocha, Gardim, Pinto, Mendonça, Ferreira Filho, Carvalho, Camargo, Oliveira, Nascimento, Cilense, Almeida, 2012) in Roraima [27], as well as the presence of *R. montenegrensis* in the state of Amazonas [28]. On the other hand, new occurrences of *Microtriatoma borbai* (Lent & Wygodzinsky, 1979) were found in Espírito Santo [29],

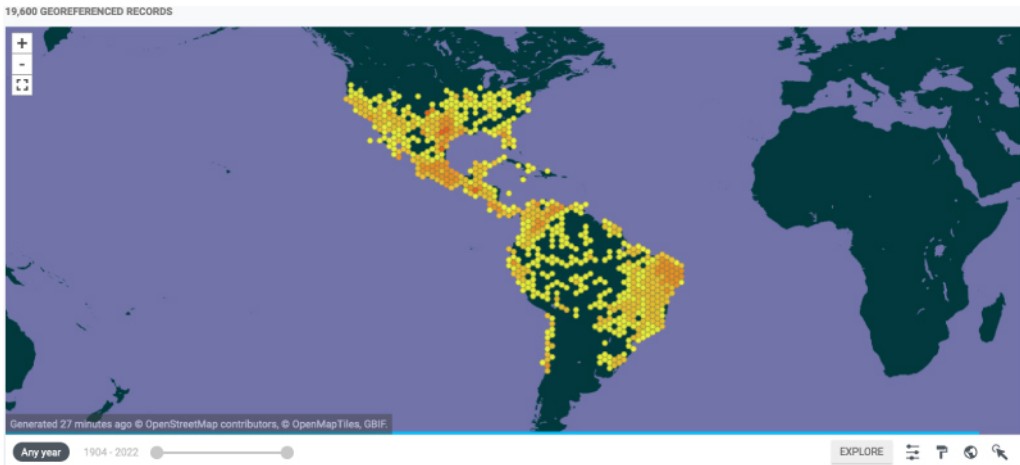

**Figure 2.** Interactive map of the georeferenced occurrences (New American dataset) hosted by GBIF [5]. https://www.gbif.org/dataset/eae731a7-3e82-4295-b0b3-ec72d75a402d

*Psammolestes tertius* (Lent & Jurberg, 1965) in the states of Sergipe [30], Paraná [31] and Rio Grande do Norte [32] in Brazil, and of *Cavernicola pilosa* and *Microtriatoma trinidadensis* in Amazonas, Brazil [33, 34].

In the New American dataset, Jamaica was included for the first time, with records of *Nesotriatoma obscura* species [35]. In summary, more than 70% of the species present in the "American dataset" now have new records incorporated in the "New American dataset" (for more details, see Table 1).

The new compiled information included in the American subdataset spans 1935 to 2022. Date information was available for 90% of the records, while 30% of them comprise data from the last 4 years. One noteworthy update to the temporal patterns of some species is that *Rhodnius pictipes* (Stål, 1872) has been distributed in Trinidad and Tobago since 1985 [36].

The addition of new records (*N* = 7852) into the New American dataset (reaching 19,600 records), plus those records included in the Argentinean dataset (*N* = 15,917) equates to a total of 35,517 occurrence records for 142 American triatomine species (Figure 2).

## METHODS

### Information source types and compilation of triatomine species data

To build the American subdataset, data for each triatomine species were obtained through a detailed and exhaustive review of information. No specific temporal range limits were set to obtain the greatest possible amount of new data from as many American countries as possible. Several public bibliographic repositories were used online (BioOne, Google Scholar, PLoS, PubMed, Scielo, ScienceDirect, Wiley) and were reviewed using terms such as "Chagas disease", "Triatominae" and "*Trypanosoma cruzi*" without language restriction. We also reviewed the public and open access triatomine bibliographic database BibTri [37]. Where published articles mentioned unpublished datasets, we contacted the authors and asked them to provide geographic coordinates, or at least locality data, to georreference them.



## Data georeferencing process

To rigorously associate each record to a specific location in the geographical space, data must have information expressed in geographic coordinates (latitude and longitude). If no geographic coordinates were available, the site name was used together with information on administrative divisions to attain an accurate location using Google Earth [38]. If the geographic coordinates were not expressed in decimal degrees, they were converted using a coordinate conversion application [39]. Where only the geographic coordinates were available, the corresponding administrative divisions were completed using GeoLoc [40]. The datum (coordinate system and set of reference points used to locate places on Earth) used for all geographic records was WGS84 (World Geodetic System 1984). The final dataset was built after data quality control.

## Description of American subdataset fields

We compiled all relevant and available information associated with each triatomine species and attached the data to each dataset field, including characteristics of the specimens collected and of the sampled sites. To better describe the fields (based on Darwin Core terms [7]) used to systematize the information, they were grouped into the following six categories: (1) identifiers (including fields used to identify each record, e.g. occurrence ID, institution code, language of the resource, associated references, etc.); (2) systematic (including fields used for systematic information, e.g. scientific name, scientific name authorship, taxon rank and taxon remarks); (3) geographical (including fields with information such as administrative divisions, coordinates, georreference sources, etc.); (4) temporal (including fields related to the event date such as year, month and day); (5) sampling (including fields related to the sampling process such as name(s) of specimen collector(s), sampled habitat, sampling protocol and effort, etc.); and (6) individual (fields related to the total number, sex and life stage of individuals sampled). The independent fields are part of the datafile mentioned in the Data Availability section. The following subsections provide some details about some of the abovementioned fields requiring specific clarifications.

### *Systematic fields*

When appropriate, the "taxonRemarks" field included notes and/or references about synonyms or formal transferals of the species described in the corresponding record.

### *Geographical fields*

The "locality" field refers to the site name nearest to the geographic coordinates – not necessarily the name of the locality where the specimens were collected. If the more accurate geographic information was the municipality name, the coordinates correspond to its centroid.

### *Temporal fields*

When a group of specimens' information corresponded to a certain time period but with specific dates available, the data were split into different records. If it was not possible to split the data, each record included the original time interval information (in years, month or days) in the "eventDate" field (e.g. 2005/2006).



### Sampling fields

The "habitat" field refers to the type of habitat where the triatomines were collected, and classified into three categories: domicile, peridomicile and sylvatic. When specific habitat information was aggregated, the habitat was expressed as a combination of two or three of those categories (e.g., domicile–peridomicile, domicile–sylvatic, peridomicile–sylvatic or domicile–peridomicile–sylvatic).

For the "SamplingProtocols" field, the available information was classified into two major categories: (i) active search, when the searching involved specialized staff; and (iii) passive collection, when different types of traps (e.g., light or Noireau traps) were used.

## DATA VALIDATION AND QUALITY CONTROL

The American subdataset was subjected to exhaustive quality control. First, each datum was extracted by one person and checked by two other people to ensure accuracy and to verify no duplication of records. Subsequently, data were checked to avoid errors (e.g., typing, georeferencing, incorrect locations, synonyms, errors in spelling of administrative divisions) that might have arisen during compilation or data entry. To correct and remove typographical errors and spelling mistakes in the names of administrative divisions, we used OpenRefine software (RRID:SCR_021305) [41], which helps to detect these types of errors in large datasets.

All geographic coordinates were checked using open GIS software (QGIS, RRID:SCR_018507 [42]) to detect georeferenced errors and incorrect locations, ensuring that each point corresponded to a location on the continent and in the correct country. Any outlier coordinates that were geographically distant from the known distribution of a given species were investigated to ensure correctness. When validating geographic coordinates, we detected that some occurrence data from public sources were located outside the continent or within continental waterbodies. These data may have been erroneously georeferenced by the authors of the original scientific publication; however, when we considered these data were sufficiently valuable to be incorporated, we carried out the following procedure: if the "country", "stateprovince", "municipality" and "locality" fields were provided by the authors, we assigned the correct geographic coordinate, taking as a reference the name of the locality contributed by the authors. To detect taxonomic synonym errors, we used the most recent triatomine review of currently valid species [2]. If any species name was suspected to be outdated, we consulted current bibliography or requested the expert opinion of colleagues.

Finally, we improved the quality of our final dataset using the GBIF data validator [43] to identify and address potential issues prior to the dataset publication through the Integrated Publishing Toolkit (IPT, [44]).

## REUSE POTENTIAL

As the information contained within the dataset has been collected using different procedures, this compilation may contain some inherent biases, which should be addressed when the data are used. Most of the data were obtained from papers published in scientific journals, accompanied by those provided by colleagues. Although data (from New American dataset + Argentinean dataset) span 25 countries, there were some countries, such as the USA, Costa Rica, Brazil and Mexico, for which the volume of data was higher than the others. In the case of the USA, the number of occurrence data included is

influenced by the great contribution of colleagues who led a kissing bug community science program [45], and are co-authors of this work (Rachel Curtis-Robles and Sarah A. Hamer). Three important notes about the data from that program are: (1) like all community science programs, records reflect locations and times where/when members of the public were aware of the program and chose to submit samples, which may not be fully representative of the true species distributions; (2) dates reflect when the sample was collected, but some samples were dead when collected and so do not reflect the phenology of the species; and (3) some samples have already been included in publications – linkages can be made using the "otherCatalogNumbers" field. In Costa Rica, an important number of records were integrated from the Instituto Nacional de Biodiversidad (INBio) and deposited in the Museo Nacional de Costa Rica, provided by a co-author of this work (José Manuel Ayala Landa).

In Mexico and Brazil, the number of occurrence data records per country included seems to be mainly influenced by two factors: (i) the number of triatomine species present in each country (both countries have the highest number of triatomine species); and (ii) the number of occurrence data records published and provided by colleagues (Mexico and Brazil are also the countries with the largest amount of data collected). An explanation for the latter factor goes beyond the goal of this paper. For habitat sampling, we recognize a potential bias in favor of the domiciliary and peridomiciliary habitats because these are the habitats of major epidemiological importance and the target of vector control campaigns. Additionally, the paucity of sylvatic habitat data also results from the difficulty of sampling procedures in the large variety of sylvatic habitats used by triatomines. Finally, it is worth noting that about 37% of the records lack available date information; thus, we recommend that any analysis based on this dataset should use methods that take such biases into account.

Despite the information biases described above, the American subdataset described in this paper, and integrated in the New American dataset, plus the complementary Argentinean dataset, constitute a valuable compilation of geographic data on American triatomines, which is as complete, updated and integrated as possible. Thus, all datasets mentioned herein better represent the number of species and countries, and have more accurate geographic coordinates. Since these datasets are hosted in an open and public repository, we hope that they will contribute towards fulfilling national and international goals, such as promoting the exchange of biological information, increasing and improving the accessibility of such information, providing biological data produced and compiled in several countries, and enhancing knowledge of both the biodiversity and epidemiological data related to Chagas disease.

## DATA AVAILABILITY

The dataset "Datos de ocurrencia de triatominos americanos del Laboratorio de Triatominos del CEPAVE (CONICET-UNLP)" has been published by Centro de Estudios Parasitológicos y de Vectores (CEPAVE) [46] and is available in the GBIF repository under a CC0 public domain waiver [5].

## EDITOR'S NOTE

This paper is part of a series of Data Release articles working with GBIF and supported by the Special Programme for Research and Training in Tropical Diseases (TDR), hosted at the World Health Organization [47].

## DECLARATIONS
## LIST OF ABBREVIATIONS

GBIF: Global Biodiversity Information Facility; IPT: integrated publishing toolkit.

## ETHICAL APPROVAL

Not applicable.

## CONSENT FOR PUBLICATION

Not applicable.

## COMPETING INTERESTS

The authors declare that they have no competing interests.

## FUNDING

This research was funded by grants PICT no. 2018-1545 to GM from Agencia Nacional de Promoción Científica y Tecnológica (ANPCyT), and PICT N° 2018-0707 to SC.

## AUTHORS' CONTRIBUTIONS

SC, AB, MEV, JER and GAM drafted the data collection protocol. SC, AB and GAM requested unpublished data from researchers. AB and MEV quality controlled the data, RCR and SAH quality controlled data from the Kissing Bug Community Science Program (USA), and JMAL quality controlled data from InBio and deposited in the Museo Nacional de Costa Rica. SC wrote the first draft and all authors contributed to the manuscript. All authors read and approved the final version of the manuscript.

## ACKNOWLEDGEMENTS

The authors are grateful to the people who provided unpublished data, and to the authors who confirmed details related to their published work and who are cited in the published datasets linked to this data paper. SC and GAM had full access to all the data in the study and take responsibility for the integrity of the data and the accuracy of the data analysis.

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
