## [Reviewer Report]

Upload additional filesDRR-202203-02/form/Data release paper_Ceccarelli et al 2022 (1).docxReviewer name and names of any other individual's who aided in reviewer Jader de OliveiraDo you understand and agree to our policy of having open and named reviews, and having your review included with the published papers. (If no, please inform the editor that you cannot review this manuscript.)YesIs the language of sufficient quality?YesPlease add additional comments on language quality to clarify if needed
Are all data available and do they match the descriptions in the paper? YesAdditional CommentsDear authors, I was very happy to see how much information was generated and updated, congratulations for the effort and the beautiful work.

My comments are to try to add and enrich the work. I inserted it as a comment in the file.

Any questions do not hesitate to write.Are the data and metadata consistent with relevant minimum information or reporting standards? See GigaDB checklists for examples <a href="http://gigadb.org/site/guide" target="_blank">http://gigadb.org/site/guide</a>YesAdditional CommentsIs the data acquisition clear, complete and methodologically sound?YesAdditional CommentsIs there sufficient detail in the methods and data-processing steps to allow reproduction?YesAdditional CommentsIs there sufficient data validation and statistical analyses of data quality? YesAdditional CommentsIs the validation suitable for this type of data?YesAdditional CommentsIs there sufficient information for others to reuse this dataset or integrate it with other data?YesAdditional CommentsAny Additional Overall Comments to the AuthorRecommendationMinor Revision

---

## [Reviewer Report]

Upload additional filesDRR-202203-02/form/DRR-202203-02_Data-Review-MAT.docx.pdfReviewer name and names of any other individual's who aided in reviewer Mary Ann TuliDo you understand and agree to our policy of having open and named reviews, and having your review included with the published papers. (If no, please inform the editor that you cannot review this manuscript.)YesIs the language of sufficient quality?NoPlease add additional comments on language quality to clarify if needed
The language is fine, but it is hard to make complete sense of as there are lots of different dates and numbers (of records) and names for the different datasets.Are all data available and do they match the descriptions in the paper? YesAdditional CommentsAre the data and metadata consistent with relevant minimum information or reporting standards? See GigaDB checklists for examples <a href="http://gigadb.org/site/guide" target="_blank">http://gigadb.org/site/guide</a>YesAdditional CommentsIs the data acquisition clear, complete and methodologically sound?NoAdditional CommentsAs above, I think it is partly a language issue, but I think this section could be made more clear to readers. Maybe splitting the section "Description of new American triatomine occurrence data" into smaller sub-sections would make it easier to read and follow. It is very rich in names, dates and numbers. Is there sufficient detail in the methods and data-processing steps to allow reproduction?YesAdditional CommentsIs there sufficient data validation and statistical analyses of data quality? YesAdditional CommentsIs the validation suitable for this type of data?YesAdditional CommentsIs there sufficient information for others to reuse this dataset or integrate it with other data?YesAdditional CommentsAny Additional Overall Comments to the AuthorI believe all the information is available, but its value is somewhat lost in the presentation. RecommendationMinor Revision